# Bandits Meet Mechanism Design
# to Combat Clickbait in Online Recommendation

**Thomas Kleine Buening** [1]   **Aadirupa Saha** [2]   **Christos Dimitrakakis** [3]   **Haifeng Xu** [4]

## Abstract

We study a strategic variant of the multi-armed bandit problem, which we coin the *strategic click-bandit*. This model is motivated by applications in online recommendation where the choice of recommended items depends on both the click-through rates and the post-click rewards. Like in classical bandits, rewards follow a fixed unknown distribution. However, we assume that the click-through rate of each arm is chosen strategically by the arm (e.g., a host on Airbnb) in order to maximize the number of times it gets clicked. The algorithm designer does not know the post-click rewards nor the arms' actions (i.e., strategically chosen click-rates) in advance, and must learn both values over time. To solve this problem, we design an incentive-aware learning algorithm, UCB-S, which achieves two goals simultaneously: (a) aligning incentives by incentivizing desirable arm actions under uncertainty; (b) learning unknown parameters. We approximately characterize all Nash equilibria among arms under UCB-S and show a $\widetilde{O}(\sqrt{KT})$ regret bound in every equilibrium. We also show that incentive-unaware algorithms generally fail to achieve low regret in the strategic click-bandit setup.

## 1. Introduction

Recommendation platforms act as intermediaries between *vendors* and *users* so as to recommend *items* from the former to the latter. On Amazon, vendors sell physical items, while on Youtube the recommended items are videos. The recommendation problem is how to select one or more items to present to each user so that they are most likely to click on at least one of them.

However, vendor-chosen *item descriptions* are an essential

aspect of the problem that is often ignored. These invite vendors to exaggerate their true value in the descriptions in order to increase their Click-Through-Rates (CTRs). As a consequence, though online learning algorithms can generally identify relevant items, the existence of unrepresentative or exaggerated item descriptions remains a challenge (Yue et al., 2010; Hofmann et al., 2012). These include thumbnails or headlines that do not truly reflect the underlying item (see Figure 1)—a well-known internet phenomenon called the *clickbait* in the context of articles. While moderately increasing user click-rates through attractive descriptions is often encouraged since it helps to increase the overall user activity, clickbait can be harmful to a platform as it leads to bad recommendation outcomes and damage to the platform's reputation which may exceed the value of any additional clicks it brings. A key reason for such dishonest or exaggerated item deceptions is the *strategic behavior* of vendors driven by their incentive to increase their item's exposure and click probability. Thus naturally, vendors are better off carefully choosing descriptions so as to increase click-rates, which leads to phenomena such as clickbait.[1]

To address this issue, we take an approach that marries *mechanism design* without payments to *online learning*, which are two celebrated research areas, however, mostly studied as separate streams. Since clickbait is fundamentally driven by vendor incentives, we believe that the novel design of online learning policies *that can carefully align vendor incentives with the platform's overall objective* may help to resolve this issue from its root.

To incorporate vendor-chosen item descriptions in this setting, we propose and study a natural strategic variant of the classical Multi-Armed Bandit (MAB) problem, which we call the *strategic click-bandit* in order to emphasize the strategic role that clicks and CTRs play in our setup.[2] Concretely, in strategic click-bandits, each arm $i$ is characterized by (a) a reward distribution with mean $\mu_i$, inherent to the

---

[1]University of Oslo [2]Apple ML Research [3]University of Neuchatel [4]University of Chicago. Correspondence to: Thomas Kleine Buening <thomkl@ifi.uio.no>.

Interactive Learning with Implicit Human Feedback Workshop at ICML 2023. Honolulu, Hawaii, USA.

[1]This is possible because most platforms rely on vendors to provide descriptions about their items. For instance, the images of restaurants on Yelp, rentals on Airbnb, hotels on Expedia, title and thumbnails of Youtube videos, and descriptions of products on Amazon are all provided by these items' vendors.

[2]We use the terms click-through-rate, click-rate, and click probability interchangeably.

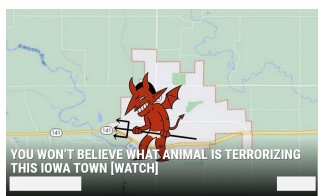 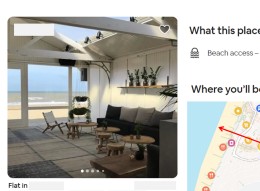 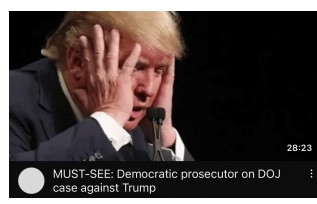 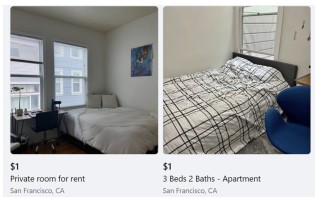

*Figure 1.* Examples of unrepresentative or clickbait headlines and thumbnails on Bing News, Airbnb, Youtube, and Facebook Marketplace (identifying information partly redacted).

arm; and (b) a click probability $s_i \in [0, 1]$, chosen freely by the arm at the beginning. Since the learner (i.e., the recommendation system) knows neither of these values, it must learn them through interaction. The learner's objective is represented through a general utility function $u(s_i, \mu_i)$ that depends on both click-rate and post-click rewards.

The strategic click-bandit proceeds in two phases. In the first phase, the learner commits to an online learning policy $M$, upon which each arm $i$ chooses a description, which results in a corresponding CTR $s_i \in [0, 1]$. The second phase proceeds in rounds. At each round $t$: (1) the algorithm $M$ pulls/recommends an arm $i_t$ based on observed past data; (2) arm $i_t$ is clicked with probability $s_{i_t}$; (3) if $i_t$ is clicked, $c_{t,i_t} = 1$, the arm $i_t$ receives utility 1 (whereas all other arms $i$ receive utility 0 and $c_{t,i} = 0$) and the learner observes a post-click reward drawn from $i_t$'s reward distribution. If $i_t$ is *not* clicked, all arms receive 0 utility and the learner does not observe any post-click rewards. From the learner's perspective, *both* $s_i$ and $\mu_i$ of each arm are unknown but can be learned from online bandit feedback — that is, whether the recommended arm is clicked and, if so, what its realized reward is.

We highlight two fundamental differences between strategic click-bandits and standard MABs. First, each arm in the strategic click-bandit is a *self-interested agent* whose objective is to maximize the number of times it gets clicked $\sum_{t=1}^{T} \mathbb{1}_{\{i_t=i\}} c_{t,i}$ with $c_{t,i} \sim \text{Bern}(s_i)$. This captures the strategic behavior of many vendors in online recommendations, especially those who are rewarded based on user clicks (e.g., Youtube (2023)). Second, $s_i \in [0, 1]$ is a freely chosen *action* by arm $i$, rather than a fixed parameter of arm $i$. We believe these modeling adjustments more realistically capture vendor behaviors in real applications. They also lead to intriguing mechanism design questions since the bandit algorithm not only needs to learn the unknown parameters $s_i, \mu_i$, but also has to carefully align arms' incentives to avoid undesired arm behavior.

In summary, our contributions are:

1. We introduce the strategic click-bandit problem, which involves strategic arms manipulating click-rates so as to maximize their own utility and show

that *incentive-unaware* algorithms generally fail to achieve low regret in the strategic click-bandit (Section 3, Proposition 4.2).

2. We design an *incentive-aware* learning algorithm, UCB-S, that combines mechanism design and online learning techniques and effectively incentivizes desirable arm strategies while minimizing regret by making credible and justified threats to arms under uncertainty (Section 5).

3. We characterize the Nash equilibria among arms under the UCB-S mechanism and show that every arm $i$'s strategy is $\widetilde{O}\big(\max\big\{\Delta_i, \sqrt{K/T}\big\}\big)$ close to the desired strategy (Theorem 5.2).

4. We show that UCB-S achieves $\widetilde{O}\big(\sqrt{KT}\big)$ strong strategic regret (Theorem 5.3) and complement this with an almost matching lower bound on weak strategic regret (Theorem 5.5).

## 2. Related Work

The MAB problem is a well-studied online learning framework, which can be used to model decision-making under uncertainty (Lai et al., 1985; Auer, 2002). Since it inherently involves sequential actions and the exploration-exploitation trade-off, the MAB framework has been applied to online recommendations (Li et al., 2010; Zong et al., 2016; Wang et al., 2017) as well as a myriad other domains (Bouneffouf et al., 2020). While there is a wide spectrum of work concerned with strategic machine learning (e.g., Hardt et al., 2016; Freeman et al., 2020; Zhang and Conitzer, 2021), we focus on related work that connects online learning (and specifically the MAB formalism) to mechanism design (Nisan and Ronen, 1999).

To the best of our knowledge, Braverman et al. (2019) is the first to study a strategic variant of the multi-armed bandit problem. In their model, when an arm is pulled, it receives a privately observed reward $\nu$ and chooses to pass on a portion $x$ of it to the principal, keeping $\nu - x$ for itself. The goal of the principal is then to incentivize arms to share as much reward with the principal as possible. In contrast to our work, the principal must not learn the underlying reward dis-

tribution or the arm strategies, but instead design an auction among arms based on the shared rewards. Feng et al. (2020) and Dong et al. (2022) study the robustness of bandit algorithms to strategic reward manipulations. However, neither work attempts to align incentives by designing mechanisms, but instead assume a limited manipulation budget. Shin et al. (2022) study multi-armed bandits with strategic replication in which agents can submit several arms with replicas to the platform. They design an algorithm, which separately explores the arms submitted by each agent and in doing so discourages agents from creating additional arms and replicas.

Another line of work studies auction-design in multi-armed bandit formalisms, often motivated by applications in ad auctions (Babaioff et al., 2009; Devanur and Kakade, 2009; Babaioff et al., 2015). In these models, in every round the auctioneer selects one advertiser's item, which is subsequently clicked or not. Each advertiser has a private value-per-click, which is unknown to the auctioneer, and instead submits a bid, which may differ from their value-per-click. The designer's goal is to incentivize agents to truthfully bid their value-per-click so as to maximize social welfare by constructing selection and payment rules. Similarly, Gao et al. (2021) study an auction-based combinatorial multi-armed bandit with payments, where each arm can misreport a cost for its selection.

To the best of our knowledge, our work is the first to study the situation where the arms' strategies (as well as other parameters) are initially unobserved, but must be learned from interaction while simultaneously incentivizing arms under uncertainty without payments. As a result, while other work is usually able to precisely incentivize certain arm strategies, our mechanism design and characterization of the Nash equilibria are *approximate*.

## 3. The Strategic Click-Bandit Problem

We consider a natural strategic variant of the classical MAB, motivated by applications in online recommendation. Unlike classical MABs, strategic click-bandits feature decentralized interactions with the learner and multiple self-interested arms. In the following, we will also refer to this online learning policy as a *mechanism* to emphasize its dual role in learning and incentive design.

Let $[K] := \{1, \ldots, K\}$ denote the set of arms, each being viewed as a strategic *agent*. At the beginning, the leaner commits to a history-dependent arm selection policy $M$, which is made public. Then, each arm chooses a strategy $s_i \in [0, 1]$, which determines the probability of the arm being clicked (i.e., the CTR) *after* it is pulled (i.e., after it is recommended by the learner). Then, at each round $t$, the learner selects an arm $i_t$. The selected arm is clicked with

probability $s_{i_t}$. If the arm is clicked, then $c_{t,i_t} = 1$ and a stochastic *reward* $r_t$ with mean $\mu_{i_t}$, is generated. This mean $\mu_i$ is fixed for each arm $i$ captures the *true value* of this arm. Because we only observe the reward when arm gets clicked, we call the setting "click-bandits". We summarize the interaction in Model 1.

---

**Model 1** The Strategic Click-Bandit Problem
1: Learner commits to algorithm $M$ (shared with all arms)
2: Arms choose strategies $(s_1, \ldots, s_K) \in [0, 1]^K$
3: **for** $t = 1, \ldots, T$ **do**
4:     Algorithm $M$ selects arm $i_t \in [K]$
5:     Arm $i_t$ is clicked w.p. $s_{i_t}$, i.e., $c_{t,i_t} \sim \mathrm{Bern}(s_{i_t})$
6:     **if** $i_t$ was clicked ($c_{t,i_t} = 1$) **then**
7:         Arm $i_t$ receives utility 1 from the click
8:         $M$ observes post-click reward $r_{t,i_t} \in [0, 1]$ sampled from a distribution with mean $\mu_i$

---

The following two assumptions specify the information available to the learner and the arms.

**Assumption 3.1.** The learner does not see the arm strategies $s_1, \ldots, s_K$ in advance, but only observes whether the arm is clicked or not in round $t$. Similarly, the learner does not know the mean post-click rewards $\mu_1, \ldots, \mu_K$ in advance, but only observes the realized reward $r_{t,i_t}$ if $i_t$ is clicked.

**Assumption 3.2.** Every arm $i \in [K]$ has private knowledge of their mean post-click reward $\mu_i$, but knows the maximal value among all arms (not index) $\mu^* := \max_{i \in [K]} \mu_i$.

### 3.1. Learner's Utility

The learner's utility of selecting an arm $i$ with CTR $s_i$ and post-click value $\mu_i$ is denoted as $u(s_i, \mu_i)$. One example of this utility function is $u(s, \mu) = s\mu$. In this case, the learner monotonically prefers large $s$ and does not care about how much the click-rate $s$ differs from the post-click value $\mu$. However, we believe that the learner (e.g., a platform like Youtube or Airbnb) usually values consistency between the click-rates and the post-click values of arms. This could be captured by a penalty term for how much $s_i$ differs from $\mu_i$; for instance, another natural choice is $u(s, \mu) = s\mu - \lambda(s - \mu)^2$ for a weight $\lambda > 0$. Such *non-monotonicity* of the learner's utility $u(s_i, \mu_i)$ in $s_i$ *versus* arm $i$'s monotonic preference of larger click-rates $s_i$ forms the fundamental tension in the strategic click-bandit model and is also the reason that mechanism design is needed. We keep the above utility functions in mind as running examples, but derive our results for a general class of functions $u$ satisfying the following regularity assumptions:

(A1) $u \colon [0, 1] \times [0, 1] \to \mathbb{R}$ is $L$-Lipschitz.

(A2) $u^*(\mu) := \max_{s \in [0,1]} u(s, \mu)$ is increasing.

(A3) $s^*(\mu) := \mathrm{argmax}_{s \in [0,1]} u(s, \mu)$ is $H$-Lipschitz.

(A1) bounds the loss of selecting a suboptimal arm. (A2) states that, in the (idealized) situation when the arms choose CTR $s$ so as to maximize the learner's utility $u$, then an arm with larger post-click rewards $\mu$ is preferred. (A3) then ensures that the ideal strategy under a utility function $u$, as a function of $\mu$, does not change abruptly w.r.t. $\mu$. From hereon-out, we work under a general learner's utility $u$ satisfying (A1)-(A3).

In the following, the function $s^*(\mu)$ will play a central role as it describes the arm strategy that maximizes the learner's utility. As such, the learner will typically try to incentivize an arm with post-click reward $\mu$ to choose $s^*(\mu)$. Finally, we remark that our analysis straightforwardly generalizes to the case in which different arms correspond to different utility functions $u_i$. However, for notational convenience, we decide to use the same $u$ among arms for the rest of the paper. Moreover, while we assume that the utility function $u$ does not depend on the arms' stochastic reward realization $r_{t,i}$, we can equivalently think of $u$ depending on $r_{t,i}$ *linearly* so that the dependency boils down to the dependence on $\mu_i$ after taking expectation. Assuming such quasi-linear utilities is standard in most literature, and is widely known as the von Neumann–Morgenstern utility assumption in economics.

### 3.2. Arms' Utility and Nash Equilibria Among Arms

The mean $\mu_i$ of each arm $i$'s true value is fixed, whereas arm $i$ can freely choose the CTR value $s_i$. In the strategic click-bandit, the objective of each arm $i$ is to maximize the number of times it gets clicked $\sum_{t=1}^T \mathbb{1}_{\{i_t=i\}} c_{t,i}$, which captures the incentives of many vendors on internet platforms, for whom user traffic typically proportionally converts to revenue.[3] We now introduce the solution concept for the game among arms defined by a mechanism $M$ and post-click rewards $\mu_1, \ldots, \mu_K$, often referred to as an *equilibrium*. Let $s_{-i}$ denote the $K-1$ strategies of all arms except $i$. Each arm $i$ chooses $s_i$ to maximize their *expected* number of clicks $v_i(M, s_i, s_{-i})$, which is a function of their own action $s_i$, the mechanism $M$ as well as all other arms' actions $s_{-i}$. Concretely,

$$v_i(M, s_i, s_{-i}) := \mathbb{E}_M \left[ \sum_{t=1}^T \mathbb{1}_{\{i_t=i\}} c_{t,i} \right] \quad (1)$$

where the expectation is taken over the mechanism's decisions and the environment's randomness. We generally write $s := (s_1, \ldots, s_K)$ to summarise a strategy profile of

---

[3]More generally, different arms $i$ may have a different value-per-click $\nu_i$ that could as well depend on $\mu_i$ so that $v_i(M, s_i, s_{-i}) = \mathbb{E}_M[\sum_t \mathbb{1}_{\{i_t=i\}} c_{t,i} \nu_i]$. This can easily be accommodated for by our model and our results readily extend to this case since each arm's goal still boils down to maximizing the number of clicks.

the arms. Let $\Sigma$ denote the set of probability measures over $[0, 1]$. For a *mixed strategy profile* $\boldsymbol{\sigma} = (\sigma_i, \sigma_{-i}) \in \Sigma^K$, i.e., a distribution over pure strategies, arm $i$'s utility under mechanism $M$ is then defined as $v_i(M, \sigma_i, \sigma_{-i}) := \mathbb{E}_{s \sim \boldsymbol{\sigma}}[v_i(M, s_i, s_{-i})]$.

**Definition 3.3** (Nash Equilibrium). We say that $\boldsymbol{\sigma} = (\sigma_1, \ldots, \sigma_K) \in \Sigma^K$ is a Nash equilibrium (NE) under mechanism $M$ if $v_i(M, \sigma_i, \sigma_{-i}) \geq v_i(M, \sigma_i', \sigma_{-i})$ for all $i \in [K]$ and strategies $\sigma_i' \in \Sigma$.

In other words, $\boldsymbol{\sigma}$ is a Nash equilibrium if no arm $i$ can increase its utility by *unilaterally* deviating to any $\sigma_i'$. If a Nash equilibrium $\boldsymbol{\sigma} \in \Sigma^K$ is a pure strategy profile $s \in [0, 1]^K$ (i.e., each $\sigma_i$ is a point distribution supported on a single action $s_i$), this equilibrium is said to be a pure-strategy Nash equilibrium. Let $\text{NE}(M) := \{\boldsymbol{\sigma} \in \Sigma^K : \boldsymbol{\sigma} \text{ is a NE under } M\}$ denote the set of all (possibly mixed) Nash equilibria for the arms under mechanism $M$. Following conventions in standard economic analysis, we assume that arms' strategies will from a Nash equilibrium in $\text{NE}(M)$.

**Remark 3.4** (Existence of Nash Equilibrium). In general, the arm's utility function $v_i(M, s_i, s_{-i})$ may be discontinuous in the arms' strategies due to its intricate dependence on the learning algorithm $M$. It is well-known that in games with discontinuous utility functions, a Nash equilibrium may not exist (Reny, 1999). Nevertheless, for all algorithms we consider we will prove the existence of a Nash equilibrium by either explicitly describing the equilibrium or implicitly proving its existence.

### 3.3. Strategic Regret

The learner's goal is to maximize $\sum_{t=1}^T u(s_{i_t}, \mu_{i_t})$, which naturally depends on the strategies $s_1, \ldots, s_K$. For given post-click values $\mu_1, \ldots, \mu_K$, the maximal utility $u(s^*, \mu^*)$ is then achieved for $\mu^* := \max_{i \in [K]} \mu_i$ and $s^* := s^*(\mu^*)$, that is, $u(s^*, \mu^*) = \max_{i \in [K]} \max_{s \in [0,1]} u(s, \mu_i)$. With $u(s^*, \mu^*)$ as a benchmark, we then define the *strategic regret* of a mechanism $M$ under a pure-strategy equilibrium $s \in \text{NE}(M)$ as

$$R_T(M, s) := \mathbb{E} \left[ \sum_{t=1}^T u(s^*, \mu^*) - u(s_{i_t}, \mu_{i_t}) \right]. \quad (2)$$

For a mixed-strategy equilibrium $\boldsymbol{\sigma} \in \text{NE}(M)$, we then accordingly define the strategic regret as

$$R_T(M, \boldsymbol{\sigma}) := \mathbb{E}_{s \sim \boldsymbol{\sigma}}[R_T(M, s)].$$

In general, there may exist several Nash equilibria for the arms under a given mechanism $M$. Then, we can consider the *strong strategic regret* of $M$ given by the regret under the worst-case Nash equilibrium:

$$R_T^+(M) := \max_{\boldsymbol{\sigma} \in \text{NE}(M)} R_T(M, \boldsymbol{\sigma}),$$

or the *weak strategic regret* given by the regret under the most favorable Nash equilibrium:

$$R_T^-(M) := \min_{\boldsymbol{\sigma} \in \mathrm{NE}(M)} R_T(M, \boldsymbol{\sigma}).$$

Naturally, $R_T^+(M) \geq R_T^-(M)$. The regret upper bound of our proposed algorithm, UCB-S, holds under any Nash equilibrium in NE(UCB-S), thereby bounding *strong strategic regret* (Theorem 5.3). On the other hand, the lower bounds we prove (Proposition 4.2 and Theorem 5.5) hold for *weak strategic regret* (and therefore also apply to its strong counterpart).

# 4. Necessity of Incentive Design: Limitations of Incentive-Unaware Oracles

We start our analysis of the strategic click-bandit problem by showing that simply finding the arm with the largest post-click reward, $\operatorname{argmax}_i \mu_i$, or largest utility, $\operatorname{argmax}_i u(s_i, \mu_i)$, is insufficient to achieve $o(T)$ *weak strategic regret*. In fact, we find that even with oracle knowledge of $\mu_1, \ldots, \mu_K$ and $s_1, \ldots, s_K$, an algorithm may suffer linear weak strategic regret if it fails to account for the arms' strategic nature. For such incentive-*unaware* oracle algorithms, we show a $\Omega(T)$ lower bound on weak strategic regret for any utility function of the learner that is not trivial to optimize, that is, when the loss of deviating from the maximal utility $u(s^*, \mu^*)$ is locally lower bounded by a "reverse Lipschitz" constant formally defined below.

**Definition 4.1** (Locally Reverse Lipschitz)**.** The utility function $u$ is locally reverse $\widetilde{L}$-Lipschitz at $(s^*, \mu^*)$ if for all $s \in [0, 1]$ and $\mu \in [0, 1]$:

$$|u(s^*, \mu^*) - u(s, \mu)| \geq \widetilde{L} \left( |s^* - s| + |\mu^* - \mu| \right).$$

For the statement of the lower bound, recall the definition of the maximal post-click reward $\mu^* := \max_i \mu_i$ and (learner's) utility-maximizing arm strategy $s^* := s^*(\mu^*)$. Moreover, let the optimality gaps in terms of post-click rewards be given by $\Delta_i := \mu^* - \mu_i$ with minimal gap $\Delta := \min_{i:\Delta_i > 0} \Delta_i$.

**Proposition 4.2.** *Let $\mu_1, \ldots, \mu_K \in [0, 1]$ with unique $i^* \in \operatorname{argmax}_{i \in [K]} \mu_i$. For any utility function $u$ that is locally reverse $\widetilde{L}$-Lipschitz at $(s^*, \mu^*)$ with $s^* \leq c$ for some $c \in [0, 1)$, the following holds:*

*(i) Let $Oracle_\mu$ be the algorithm with oracle knowledge of $\mu_1, \ldots, \mu_K$ that plays $i_t = \operatorname{argmax}_{i \in [K]} \mu_i$ in every round $t$.*

*For every equilibrium $\boldsymbol{\sigma} \in \mathrm{NE}(Oracle_\mu)$ among the arms, the algorithm suffers regret $\Omega(\widetilde{L}T)$, i.e., its weak strategic regret is lower bounded as*

$$R_T^-(Oracle_\mu) = \Omega(\widetilde{L}T).$$

*(ii) Let $Oracle_{s,\mu}$ be the algorithm with oracle knowledge of $\mu_1, \ldots, \mu_K$ and $s_1, \ldots, s_K$ that plays $i_t = \operatorname{argmax}_{i \in [K]} u(s_i, \mu_i)$ in every round $t$ with ties broken in favor of the larger $\mu$.*

*For every equilibrium $\boldsymbol{\sigma} \in \mathrm{NE}(Oracle_{s,\mu})$ of the arms, the algorithm suffers regret $\Omega(\Delta\widetilde{L}T)$, i.e., its weak strategic regret is lower bounded as*

$$R_T^-(Oracle_{s,\mu}) = \Omega(\Delta\widetilde{L}T).$$

*Proof Sketch. (i)*: We show that, under the described algorithm, $s = 1$ is a strictly dominant strategy for arm $i^*$. This implies that arm $i^*$ plays $s_{i^*} = 1$ with probability one in every Nash equilibrium under the oracle algorithm. The claimed lower bound then follows from bounding the instantaneous regret from below using $s^*(\mu_{i^*}) \leq c$.

*(ii)*: Let $j^* \in \operatorname{argmax}_{i \neq i^*} \mu_i$. It can be seen that in any Nash equilibrium, arm $i^*$ will play the largest $s \in [0, 1]$ such that $u(s, \mu_{i^*}) \geq u(s_{j^*}, \mu_{j^*})$. We then show that (at least) either $s_{i^*} = 1$ or $u(s_{i^*}, \mu_{i^*}) = u(s^*(\mu_{j^*}), \mu_{j^*})$. Once again this allows us to bound the instantaneous regret for every round from below, which yields the claimed lower bound. $\square$

**Remark 4.3.** Interestingly, when the algorithm, $Oracle_{s,\mu}$, from Proposition 4.2 *(ii)* does not break ties in favor of the larger $\mu$ but instead uniformly at random, it can be shown that in all but a few problem instances no Nash equilibrium for the arms exists. However, for any $\varepsilon > 0$ we can explicitly construct an $\varepsilon$-Nash equilibrium for the arms under which the algorithm suffers $\Omega(\Delta\widetilde{L}T)$ regret.

# 5. No-Regret Incentive-Aware Learning: UCB-S

The results of Proposition 4.2 suggest that an incentive-unaware learning algorithm that is oblivious to the strategic nature of the arms will generally fail to achieve low regret. In particular, "unconditional" selection of any arm will likely result in undesirable Nash equilibria among arms. For these reasons, we deploy a conceptually simple screening idea, which threatens arms with elimination when deviating from the desired strategies. The challenge is that the arm strategies $s_1, \ldots, s_K$ are unknown to the mechanism ahead of time and must be learned through repeated interaction.

Let denote $n_t(i)$ be the number of times up to (and including) round $t$ that arm $i$ was selected by the learner, whereas we let $m_t(i)$ denote the number of times post-click rewards were observed for arm $i$ up to (and including) round $t$. Let $\widehat{s}_i^t$ be the average observed click-rate and $\widehat{\mu}_i^t$ the average observed post-click reward for arm $i$. We then define the

---

**Algorithm 1** UCB with Screeing (UCB-S)
___

1: **initialize:** $A_0 = [K]$
2: **for** $t = 1, \ldots, T$ **do**
3:    **if** $A_{t-1} \neq \emptyset$ **then**
4:       Select $i_t \in \text{argmax}_{i \in A_{t-1}} \overline{\mu}_i^{t-1}$
5:    **else**
6:       Select $i_t$ uniformly at random from $[K]$
7:    Arm $i_t$ is clicked with probability $s_{i_t}$, i.e., $c_{t,i_t} \sim \text{Bern}(s_{i_t})$
8:    **if** $i_t$ was clicked ($c_{t,i_t} = 1$) **then**
9:       Observe post-click reward $r_{t,i_t} \in [0,1]$
10:    **if** $\overline{s}_{i_t}^t < \min_{\mu \in [\underline{\mu}_{i_t}^t, \overline{\mu}_{i_t}^t]} s^*(\mu)$ or $\underline{s}_{i_t}^t > \max_{\mu \in [\underline{\mu}_{i_t}^t, \overline{\mu}_{i_t}^t]} s^*(\mu)$ **then**
11:       Ignore arm $i_t$ in future rounds: $A_t \leftarrow A_{t-1} \setminus \{i_t\}$

---

optimistic and pessimistic estimates of $s_i$ and $\mu_i$ as

$$\underline{s}_i^t = \widehat{s}_i^t - \sqrt{\frac{2\log(T)}{n_t(i)}}, \quad \overline{s}_i^t = \widehat{s}_i^t + \sqrt{\frac{2\log(T)}{n_t(i)}},$$

$$\underline{\mu}_i^t = \widehat{\mu}_i^t - \sqrt{\frac{2\log(T)}{m_t(i)}}, \quad \overline{\mu}_i^t = \widehat{\mu}_i^t + \sqrt{\frac{2\log(T)}{m_t(i)}}.$$

where $\underline{s}_i^t = -\infty$ and $\overline{s}_i^t = +\infty$ for $n_t(i) = 0$ and $\underline{\mu}_i^t = -\infty$ and $\overline{\mu}_i^t = +\infty$ for $m_t(i) = 0$.

In every round, UCB-S selects arms optimistically according to their post-click rewards and subsequently observes if the arm is clicked, i.e., $c_{t,i_t}$, and, if so, a post-click reward $r_{t,i_t}$. However, if an arm's click-rate $s_i$ is detected to be different from the learner's desired arm strategy $s^*(\mu_i)$, the arm is eliminated forever, expressed by the screening rule in line 1:

$$\overline{s}_{i_t}^t < \min_{\mu \in [\underline{\mu}_{i_t}^t, \overline{\mu}_{i_t}^t]} s^*(\mu) \quad \text{or} \quad \underline{s}_{i_t}^t > \max_{\mu \in [\underline{\mu}_{i_t}^t, \overline{\mu}_{i_t}^t]} s^*(\mu).$$

The only exception is when all arms have been eliminated. Then, UCB-S plays them all uniformly for the remaining rounds. To ensure that the elimination of an arm is credible and justified with high probability, we use high confidence bounds on $s_i$ and $\mu_i$. More precisely, if an arm chooses $s_i = s^*(\mu_i)$ (as asked), then with probability $1 - 1/T^2$ it will not be eliminated by the screening rule.

As a prelude to the analysis of the UCB-S mechanism, we begin by showing that there always exists a Nash equilibrium among the arms under UCB-S. As mentioned briefly in Section 3, the existence of a Nash equilibrium among the arms is not guaranteed under an arbitrary mechanism due to the arms' continuous strategy space and possibly discontinuous utility function.

**Lemma 5.1.** *For any post-click rewards $\mu_1, \ldots, \mu_K$, there always exists a (possibly mixed) Nash equilibrium for the arms under the UCB-S mechanism.*

### 5.1. Characterizing the Nash Equilibria under UCB-S

We now approximately characterize the Nash equilibria for the arms under the UCB-S mechanism. In order to prove a regret upper bound for UCB-S, it will be key to ensure that each arm $i$ plays a strategy $s_i$ which is sufficiently close to the desired strategy $s^*(\mu_i)$ (i.e., the strategy that maximizes the learner's utility). This is particularly important for arms $i^*$ with maximal post-click rewards $\mu_{i^*} = \mu^*$. If such arms $i^*$ deviate substantially from $s^*(\mu_{i^*})$, e.g., by a constant amount, the learner would be forced to suffer constant regret even when selecting arms with maximal post-click rewards.

In the following, we show that under the UCB-S mechanism every Nash equilibrium consists of strategies such that the strategies of arms with maximal post-click rewards deviate from the desired strategies by at most $\widetilde{O}(\sqrt{K/T})$. We then also show that for suboptimal arms the difference between each arm's strategy $s_i$ and the desired strategy $s^*(\mu_i)$ is governed by their optimality gap in terms of post-click rewards, given by $\Delta_i := \mu^* - \mu_i$. Recall that $H$ denotes the Lipschitz constant of $s^*(\mu)$.

**Theorem 5.2.** *For all $s \in \text{supp}(\sigma)$ with $\sigma \in \text{NE}(UCB\text{-}S)$ and all $i \in [K]$:*

$$s_i = s^*(\mu_i) + O\left(H \cdot \max\left\{\Delta_i, \sqrt{\frac{K\log(T)}{T\, s^*(\mu_i)^2}}\right\}\right).$$

*In particular, for all arms $i^* \in [K]$ with $\Delta_{i^*} = 0$:*

$$s_{i^*} = s^*(\mu_{i^*}) + O\left(H\sqrt{\frac{K\log(T)}{T\, s^*(\mu_{i^*})^2}}\right).$$

*Proof Sketch.* To characterize the strategy profiles in the support $\text{NE}(UCB\text{-}S)$, we are going to rely on the best response property of the Nash equilibrium. More precisely, for any $s \in \text{supp}(\sigma)$ with $\sigma \in \text{NE}(UCB\text{-}S)$, arm $i$'s strategy $s_i$ must be a best response to $\sigma_{-i}$, i.e., for all $s_i' \in [0,1]$:

$$v_i(\text{UCB-S}, s_i, \sigma_{-i}) \geq v_i(\text{UCB-S}, s_i', \sigma_{-i}).$$

Recall that arm $i$'s utility can be written as $v_i(\text{UCB-S}, s_i, \sigma_{-i}) = \mathbb{E}_{(s_i, \sigma_{-i})}[n_T(i)]s_i$. First, we show that $s_i \geq s^*(\mu_i)$ based on fundamental properties of UCB-S and the click-bandit model. We then proceed by establishing upper and lower bounds on $\mathbb{E}_{(s_i, \sigma_{-i})}[n_T(i)]$ under UCB-S. To this end, let $\tau_i$ be the first round that arm $i$ is not in the active set $A_t$ anymore and $\tau$ be the first round that $A_t$ is empty (and by convention $\tau_i = T$ and $\tau = T$ if $i \in A_T$ and $A_T \neq \emptyset$, respectively):

$$\tau_i := \min\{t \in [T] : i \notin A_t\}, \ \tau := \min\{t \in [T] : A_t = \emptyset\}.$$

We then split $\mathbb{E}_{(s_i, \sigma_{-i})}[n_T(i)]$ into the rounds before $\tau_i$ and after $\tau$. Note that arm $i$ is never played in the rounds between $\tau_i$ and $\tau$, since it has been eliminated from $A_t$, but other arms are still active. However, after round $\tau$ arm $i$ is played with probability $1/K$ by UCB-S.

$$\mathbb{E}_{(s_i, \sigma_{-i})}[n_T(i)] = \mathbb{E}_{(s_i, \sigma_{-i})}[n_{\tau_i}(i)] + \mathbb{E}_{(s_i, \sigma_{-i})}\left[\frac{T - \tau}{K}\right].$$

From the screening rule, we obtain the upper bound

$$\mathbb{E}_{(s_i, \sigma_{-i})}[n_{\tau_i}(i)] \leq O\left(\frac{H^2 \log(T)}{s_i(s_i - s^*(\mu_i))^2}\right).$$

Moreover, from the fact that $s_i$ must be best response to $\sigma_{-i}$ we then also find that $\mathbb{E}_{(s_i, \sigma_{-i})}\left[\frac{T-\tau}{K}\right] = O\left(\frac{1}{K}\right)$. Conversely, the UCB-type selection rule implies a lower bound

$$\mathbb{E}_{(s_i, \sigma_{-i})}[n_T(i)] \geq \Omega\left(\min\left\{\frac{\log(T)}{s_i \Delta_i^2}, s^*(\mu_i)\frac{T}{K}\right\}\right).$$

As a result, when comparing the upper and lower bound on $\mathbb{E}_{(s_i, \sigma_{-i})}[n_T(i)]$, any $s_i \in \text{supp}(\sigma_i)$ must satisfy

$$\Omega\left(\min\left\{\frac{\log(T)}{s_i \Delta_i^2}, s^*(\mu_i)\frac{T}{K}\right\}\right) \leq O\left(\frac{H^2 \log(T)}{s_i(s_i - s^*(\mu_i))^2}\right).$$

Rearranging terms then yields the claimed result, where we use that $s_i \geq s^*(\mu_i)$ as established earlier.

$\square$

## 5.2. Upper Bound on Strong Strategic Regret of UCB-S

With the Nash equilibrium characterization from Theorem 5.2 at our disposal, we are ready to prove a regret upper bound for UCB-S. We show that the *strong strategic regret* of the UCB-S mechanism is upper bounded by $\tilde{O}(\sqrt{KT})$, that is, for any $\sigma \in \text{NE}(\text{UCB-S})$ the bound holds.

**Theorem 5.3.** *Let $\Delta_i := \mu^* - \mu_i$ and let $L$ and $H$ denote the Lipschitz constants of $u(s, \mu)$ and $s^*(\mu)$, respectively. The strong strategic regret of UCB-S is bounded as*

$$R_T^+(\text{UCB-S})$$
$$= O\left(\frac{LH}{s^*(\mu^*)}\sqrt{KT \log(T)} + \frac{LH}{s^*(\mu_i)}\sum_{i:\Delta_i > 0}\frac{\log(T)}{\Delta_i}\right).$$
(3)

*In other words, the above regret bound is achieved under any Nash equilibrium $\sigma \in \text{NE}(\text{UCB-S})$.*

*Proof Sketch.* As suggested by the regret bound there are two sources of regret. Broadly speaking, the first term on the right hand side of (3) corresponds to the regret UCB-S suffers due to arms with maximal post-click rewards (i.e., $\Delta_i = 0$) deviating from the utility-maximizing strategy $s^*(\mu^*)$. For such arms Theorem 5.2 bounded the deviation by a term of order $\sqrt{K/T}$, thereby leading to the claimed $\sqrt{KT}$ bound. The second term in (3) corresponds to the regret suffered from playing arms with suboptimal post-click rewards, i.e., $\Delta_i > 0$. Using a standard UCB argument, the Lipschitzness of $u(s, \mu)$ and $s^*(\mu)$, and again Theorem 5.2 applied to $|s^*(\mu^*) - s_i| \leq |s^*(\mu^*) - s^*(\mu_i)| + O(H\Delta_i) \leq H\Delta_i + O(H\Delta_i)$ yields the claimed upper bound. $\square$

Similarly, to classical multi-armed bandits we can state a regret bound independent of the instance-dependent quantities $\Delta_i$ and translate Theorem 5.3 into a minimax-type guarantee.

**Corollary 5.4.** *The strong strategic regret of UCB-S is bounded as*

$$R_T^+(\text{UCB-S}) = O\left(\sum_{i \in [K]}\frac{LH}{s^*(\mu_i)}\sqrt{KT \log(T)}\right).$$

*Letting $s_{\min} := \min_{i \in [K]} s^*(\mu_i)$, this bound can be further refined to*

$$R_T^+(\text{UCB-S}) = O\left(\frac{LH}{s_{\min}}\sqrt{KT \log(T)}\right).$$

*In other words, the above regret bounds are achieved under any Nash equilibrium $\sigma \in \text{NE}(\text{UCB-S})$.*

In Corollary 5.4 we see that there remains a dependence on the learner's utility function $u$ and post-click rewards $\mu_i$ in the form of $s^*(\mu_i)$. The intuition for this can be explained as follows: Suppose that $s^*(\mu)$ is very small. Now, UCB-S incentivizes arms to choose strategies close to $s^*(\mu_i)$, which in turn makes learning about the post-click rewards more difficult. Hence, the incentivized strategies, while maximizing $u$, make learning about the arms' rewards harder. However, we want to stress that the utility function $u$ is chosen at the discretion of the learner and a reasonable choice of $u$ would, for instance, ensure that the incentivized click-rate is always at least $s^*(\mu) \geq 0.1$ for all $\mu$.

## 5.3. Lower Bound on Weak Strategic Regret

Complementing our regret analysis, we now show a lower bound for *weak strategic regret* in the strategic click-bandit. By definition, weak strategic regret lower bounds its strong counterpart so that the shown regret lower bound directly applies to strong strategic regret as well.

**Theorem 5.5.** *Let $M$ be any mechanism with $\mathrm{NE}(M) \neq \emptyset$. There exists a utility function $u$ satisfying (A1)-(A3) and post-click rewards $\mu_1, \ldots, \mu_K$ such that for all $\boldsymbol{\sigma} \in \mathrm{NE}(M)$:*

$$R_T(M, \boldsymbol{\sigma}) = \Omega\big(\sqrt{KT}\big).$$

*In other words, $R_T^-(M) = \Omega\big(\sqrt{KT}\big)$.*

*Proof Sketch.* Consider the utility function $u(s, \mu)$. Intuitively, for any low regret mechanism $M$ the Nash equilibrium for the arms will be in $(s_1, \ldots, s_K) = (1, \ldots, 1)$ as these strategies maximize the learner's utility $u$ and are to the advantage of the arms. In this case, the learning problem reduces to a classical multi-armed bandit and we inherit the well-known minimax $\sqrt{KT}$ lower bound. However, it is not directly clear that there exists no better mechanism that would, for instance, incentivize arm strategies $(s_1, \ldots, s_{i^*}, \ldots, s_K) = (0, \ldots, 1, \ldots, 0)$ under which $i^* = \arg\max_i \mu_i$ may become easier to distinguish from $i \neq i^*$. For this reason, we argue via the arms' utilities and lower bound the minimal utility a suboptimal arm must receive under any Nash equilibrium. Since this lower bounds the number of times we play a suboptimal arm, we recover the claimed lower bound. $\qquad \square$

## 6. Discussion

We study the strategic click-bandit problem in which each arm is associated with a click-rate, chosen strategically by the arms, and a more sparsely observed and immutable post-click reward. We show the necessity of incentive design in this model and design an incentive-aware online learning algorithm that incentivizes desirable arm strategies under uncertainty. As the learner has no prior knowledge of the arm strategies and the post-click rewards, the mechanism design is approximate and leaves room for arms to exploit the learner's uncertainty. This leads to an interesting regret bound which makes the intuition precise that arms can exploit the learner's uncertainty about their strategies.

A natural extension to the studied setting would be to assume that CTRs are user-dependent or more generally dependent on contextual information. Another direction would be to consider multi-slot recommendations in which the learner selects a subset of arms every round and the selected arms compete for the click (and our observations are hence preference-based). In fact, the case where the learner selects a set of arms and each arm $i$ is clicked with probability $s_i$ independently of the other arms can be handled with exactly the same methods as presented in this paper. In the current setup, it would also be appealing to construct online learning mechanisms under which there exists a Nash equilibrium in dominant strategies with similarly benign properties as shown in this paper.

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
