# OpenReview forum: "Bandits Meet Mechanism Design to Combat Clickbait in Online Recommendation"
_ICML.cc/2023/Workshop/ILHF — ILHF Workshop ICML 2023_

### Official Review · Reviewer_rpSf · 2023-06-14
**Review of Bandits Meet Mechanism Design to Combat Clickbait in Online Recommendation**

**Rating:** 7
**Confidence:** 3

**Review:**

This paper studies mechanism design and online learning simultaneously.  In this strategic click-bandit problem the bandit, at the beginning of the game, each arm chooses a  CTR (strategy) that is unknown to the learner. Each arm is again associated with a private mean reward(which it knows) but does not know the maximum mean of other arms in the game. The learner does not know the reward mean of the arms either. Then the goal of the learner is to minimize the regret based on a known utility function which depends on the strategy, and reward means. They provide regret upper and lower bounds under a strategy that satisfies Nash Equilibrium.

Strengths:
1) I like the formulation of this problem. It is a very relevant problem.
2) They analyze the regret upper and lower bound and clearly specify the assumptions of the setting.

Weakness:
1) The conditions on the utility function seem to be too contrived. It needs more justification.
2) It would be great to see some experiments.

I did not go too much into the theory details, but overall satisfied with their approach and have seen no glaring issues. Overall the paper packs sufficient materials to be accepted into the workshop.

---

### Decision · Program_Chairs · 2023-06-20

Accept